# Trait phenomenological control predicts experience of mirror synaesthesia and the rubber hand illusion

P. Lush [1,2✉], V. Botan[1,3], R. B. Scott[1,3], A. K. Seth [1,2,4], J. Ward[1,3] & Z. Dienes[1,3]

In hypnotic responding, expectancies arising from imaginative suggestion drive striking experiential changes (e.g., hallucinations) — which are experienced as involuntary — according to a normally distributed and stable trait ability (hypnotisability). Such experiences can be triggered by implicit suggestion and occur outside the hypnotic context. In large sample studies (of 156, 404 and 353 participants), we report substantial relationships between hypnotisability and experimental measures of experiential change in mirror-sensory synaesthesia and the rubber hand illusion comparable to relationships between hypnotisability and individual hypnosis scale items. The control of phenomenology to meet expectancies arising from perceived task requirements can account for experiential change in psychological experiments.

[1] Sackler Centre for Consciousness Science, University of Sussex, Falmer BN1 9RH, UK. [2] Department of Informatics, Chichester Building, University of Sussex, Falmer BN1 9RH, UK. [3] School of Psychology, Pevensey Building, University of Sussex, Falmer BN1 9RH, UK. [4] Canadian Institute for Advanced Research (CIFAR) Program on Brain, Mind, and Consciousness, Toronto, ON M5G 1M1, Canada. ✉email: p.lush@sussex.ac.uk

Hypnotisability scales measure individual differences in the ability to generate experience in response to imaginative suggestion within a situation designated as hypnotic. Such experiences include visual, auditory, or tactile hallucinations, pain, amnesia, and apparently involuntary motor actions. Response to hypnotic suggestion requires the ability to experience a wide variety of imagined events as real[1] and to experience a sense of involuntariness over the response[2]. Such responding therefore requires the top-down control of perception to meet expectancies arising from imaginative suggestions[3,4].

Hypnotic responding is distinct from social compliance or social desirability, and is not attributable to mere compliance[5,6]. Hypnotisability is a normally distributed and stable trait[7]. It has long been known that response to imaginative suggestion does not require hypnosis[8]. Indeed, hypnotic induction provides only a small boost in response (around 10%) compared to imaginative suggestion without induction[9], and this boost appears to be attributable to the word hypnosis[10]. Hypnosis can therefore be considered a particular context in which people engage context-general abilities to respond to imaginative suggestion (imaginative suggestibility)[11,12]. In sum, response to imaginative suggestion involves the top-down voluntary control of action and experience (which is experienced as involuntary) according to a stable trait ability, and hypnosis procedures are not necessary for successful responding; hypnosis is just one context within which imaginative suggestion effects occur[13].

There is agreement among hypnosis researchers that response to imaginative suggestion involves top-down control. For example, in *The Oxford Handbook of Hypnosis*[14], all five chapters in the section on theoretical perspectives involve control as a key part of the theories described. So our position is not a minority one; it is just accepting the current predominance of evidence and approach in the field of imaginative and hypnotic suggestion (but see Kirsch[15] for the counter proposal that response to imaginative suggestion is directly caused by expectancies). Because the term 'imaginative suggestibility' evokes unwarranted association with other forms of suggestibility, we here refer to context-general top-down effects on perception in response to imaginative suggestion as 'phenomenological control'. See Dienes et al.[16] for detailed discussion of the concept of phenomenological control in response to imaginative suggestion.

Although hypnosis is not required for imaginative suggestion effects, with a few notable exceptions[17], scales which measure response to imaginative suggestion have focused on the hypnotic context, in which suggestion is generally direct and explicit. However, phenomenological control is not restricted to response to direct suggestion. Indeed, hypnosis developed from Mesmerism, an 18th century suggestion effect in which subjects responded to implicit suggestion in a non-hypnotic context with, for example, apparently involuntary convulsions[18]. There is evidence that demand characteristics can drive experience in scientific experiments[19–23]. Although this proposal is consistent with many theoretical accounts of hypnosis and has potentially wide-ranging implications[16], it has not yet been directly investigated.

In short, perceived task requirements may drive real experience according to reliable trait differences in the ability to control phenomenology to meet expectancies (phenomenological control). As an initial test of predictions arising from this theory, we investigated mirror synaesthesia and the rubber hand illusion, candidate effects commonly used to study embodiment which, like hypnotic suggestion effects, involve striking and apparently involuntary experiential change. These studies are intended as test cases which, because of commonality with phenomenological control effects, have the greatest likelihood of providing evidence consistent with the theory.

In mirror-sensory synaesthesia, visual stimuli elicit reports of tactile sensations (mirror touch synaesthesia)[24,25] or pain (vicarious pain perception or mirror pain)[26]. Around 2% of the population may be mirror touch synaesthetes[27]. Grice-Jackson et al.[28] identify two groups of mirror pain responders: sensory localisers (19%), who report a localised, sensory experience, and affective general responders (12%), who report a generalised and emotional vicarious pain experience. These effects have been attributed to the activity of a mirror neuron system[29], when simulation of observed action in somatosensory mirroring systems overcomes a threshold of tactile awareness[30]. An alternative theory proposes that vicarious experiences of observed stimulation are creative forms of mental imagery which are 'subintentional' (e.g., fidgeting or tapping along to music) rather than intentional[31]; that is, they are not reflexive responses, but can occur without conscious awareness of an intention.

In the rubber hand illusion (RHI), the subject's hand is placed out of view and brushed at the same time as a visible fake hand, leading to subjective reports of experienced mis-location of the participant's hand and ownership over the rubber hand[32]. The RHI is typically measured in two ways, agreement or disagreement with illusion statements following brush stroke induction (subjective report) and changes in perceived hand position before and after induction (proprioceptive drift). The RHI is thought to reflect the role of multimodal integration in the conscious experience of ownership and location of one's body and supports an extensive literature on the sense of ownership. However, opinions differ as to whether the illusion requires top-down processes[33] or whether multimodal integration alone is sufficient[34].

Typically, RHI researchers measure subjective reports following both synchronous stimulation and a control condition of asynchronous stimulation. Because asynchronous induction is not expected to generate changes in experience, asynchronous condition scores are interpreted as indicating suggestion or compliance. Similarly, statements reflecting experiences which experimenters do not expect (e.g., visual hallucination) are often employed to control for suggestion and compliance. However, because these statements consistently produce agreement in a substantial proportion of participants, some researchers regard them as valid illusion measures[35]. These control methods are ineffective, as they overlook the role of expectancies in suggestion effects, and therefore existing claims that reports of RHI experience are not suggestion effects are invalid[36].

In sum, we find substantial relationships between trait hypnotisability and measures of mirror synaesthesia and the rubber hand illusion, supporting predictions of the theory that demand characteristics can drive experience and that these effects are driven by the control of phenomenology to meet task expectancies according to a stable trait ability.

## Results

**Study 1: mirror-sensory synaesthesia.** We report correlational analyses of mirror-sensory synaesthesia measures (response to video stimuli, e.g., of touch or of apparently painful experience) and subjective SWASH hypnotisability. If measures of mirror-sensory synaesthesia reflect imaginative suggestion effects driven by phenomenological control, response should correlate with hypnotisability.

Mean subjective scale SWASH hypnotisability score (average response to ten items/5) for the sample given the vicarious pain questionnaire (VPQ; $n = 404$) was 1.7 (SD = 0.8). On the VPQ itself, the mean total pain response (0–14) was 3.9 (SD = 3.8) and mean pain intensity (0–10) was 1.0 (SD = 1.3). Mean subjective scale SWASH score for the mirror touch synaesthesia (MTS; $n =$

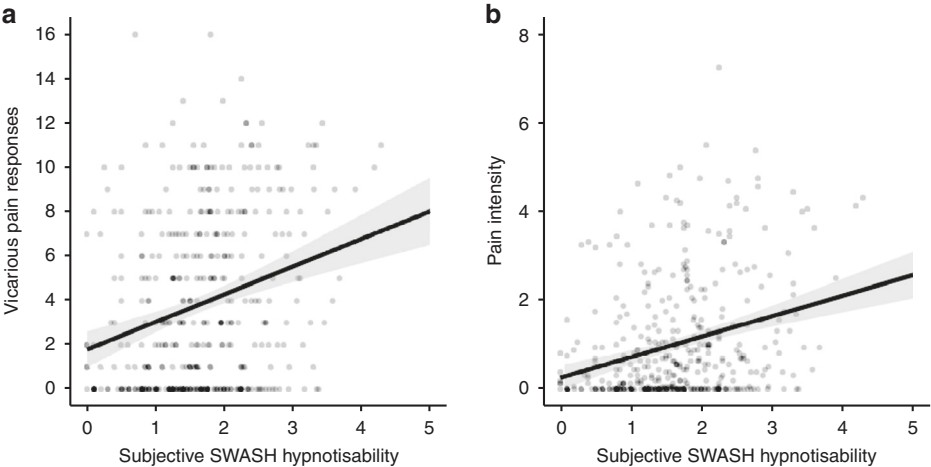

**Fig. 1 Scatter plots showing linear regression of vicarious pain measures (*n* = 404 participants) on hypnotisability. a** Total pain responses on hypnotisability, Spearman's $r_s$ = 0.26, 95% CI [0.17, 0.35]. **b** Pain intensity on hypnotisability, Spearman's $r_s$ = 0.27, 95% CI [0.18, 0.36]. Error bars show 95% CI. The centre of credibility intervals is the predicted score. Note: SWASH is an abbreviation of Sussex-Waterloo Scale of Hypnotisability.

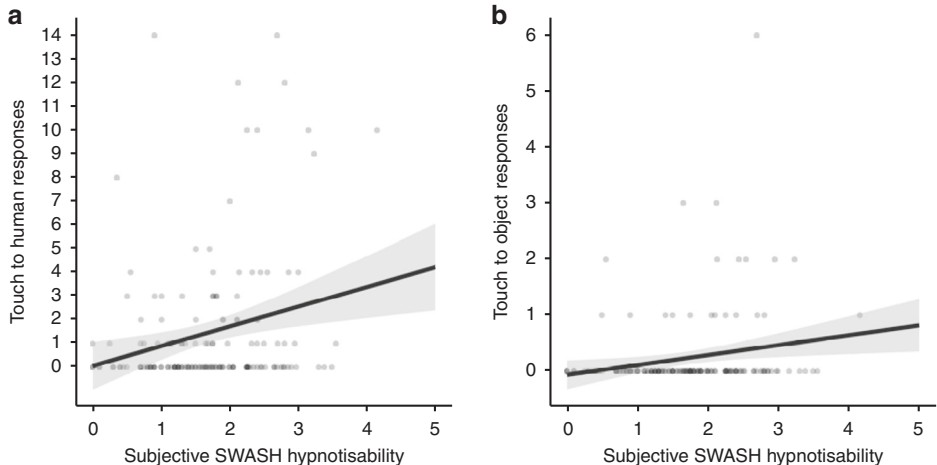

**Fig. 2 Scatter plots showing linear regression of mirror touch measures on hypnotisability (*n* = 154 participants). a** Touch to human responses, Spearman's $r_s$ = 0.19, 95% CI [0.04, 0.34]. **b** Touch to object responses, Spearman's $r_s$ = 0.21, 95% CI [0.06, 0.36]. Error bars show 95% CI. The centre of credibility intervals is the predicted score. Note: SWASH is an abbreviation of Sussex-Waterloo Scale of Hypnotisability.

154) sample was 1.7 (SD = 0.8). In this sample, mean touch to human responses was 1.5 (SD = 2.9) and touch to object responses was 0.2 (SD = 0.7).

Hypnotisability predicted VPQ total pain response (Fig. 1a), *b* = 1.25 responses/unit SWASH (SE = 0.23), *t*(402) = 5.55, *p* < 0.001, 95% CI [0.8, 1.7], $R^2$ = 0.071, $B_{H(0,1.1)}$ = 5.72 × 10$^5$, RR = [0.047, 435] and the intensity (1–10) of vicarious pain (Fig. 1b), *b* = 0.46, intensity/unit SWASH (SE = 0.08), *t*(402) = 6.00, *p* < 0.001, 95% CI [0.31, 0.62], $R^2$ = 0.082, $B_{H(0,0.3)}$ = 2.60 × 10$^6$, RR = [0.016, 163]. Regression analysis of mirror touch scores showed that hypnotisability predicted mean number of responses (out of 14) when observing touch to a human (Fig. 2a), *b* = 0.83 responses/unit SWASH (SE = 0.27), *t*(152) = 3.07, *p* = 0.003, 95% CI [0.30, 1.37], $R^2$ = 0.058, $B_{H(0,0.5)}$ = 11.98, RR = [0.011, 2.02] and number of responses (out of 6) for touch to an inanimate object (Fig. 2b), *b* = 0.18, responses/unit SWASH (SE = 0.07), *t*(152) = 2.55, *p* = 0.012, 95% CI [0.04, 0.32], $R^2$ = 0.041, $B_{H(0,0.07)}$ = 5.23, RR = [0.004, 0.12].

**Study 2: The rubber hand illusion.** Separate groups of participants completed a RHI procedure after being informed either that they should expect the illusion for synchronous induction (*n* =

114) or that they should expect the illusion for asynchronous induction (*n* = 115). A third control group was not given information about what to expect (*n* = 124). We report analyses in two sections: pre-registered analyses (intended to test the effect of experimentally manipulated expectancies on RHI measures) and non-registered analyses. Non-registered correlational analyses were chosen to reflect the most commonly reported measures of the RHI and the simplest predictions of the theory that the RHI is at least partially driven by phenomenological control.

Mean subjective scale SWASH hypnotisability score (0–5) was 1.6 (SD = 0.7) in the control group, 1.6 (SD = 0.8) for synchronous instruction and 1.7 (SD = 0.7) for asynchronous instruction.

Our pre-registered analyses included a test of whether changes in expectancies affected RHI measures. In order to test the effects of manipulating expectancies on RHI measures, we first had to check that we had successfully manipulated expectancies. Mean expectancies for synchronous induction were 0.9 (SD = 1.1) for the synchronous instruction group, 0.9 (SD = 1.1) for the asynchronous induction group and 0.7 (SD = 1.1) for the control group. Mean expectancies for asynchronous induction were −0.1 (SD = 1.2) for the synchronous instruction group, 0.2 (SD = 1.3)

for the asynchronous induction group and $-0.6$ (SD $= 1.2$) for the control group. There was no sensitive evidence for whether or not the interaction of instruction (synchronous or asynchronous) and induction condition (synchronous or asynchronous) was greater than zero on expectancies, $F(1,255) = 2.06$, $p = 0.153$, $\eta^2_\text{p}$ $= 0.009$, $B_{\text{H}(0,1.2)} = 0.72$. Therefore, our experimental manipulation of expectancy was not shown to be effective and pre-registered analyses based on differences between the suggested synchronous and suggested asynchronous conditions could not be conducted.

We next tested the pre-registered key test that control group RHI response was lower for asynchronous induction than synchronous induction (a typical control in RHI studies). Subjective report was recorded on a 7 point scale from $-3$ to $+3$, with $-3$ indicating strong disagreement and $+3$ strong agreement with a given statement. Proprioceptive drift was given by the distance in cm between pre- and post-induction reports of hand location. For controls, proprioceptive drift was greater following synchronous induction ($M = 0.93$ cm, SD $= 2.64$) than asynchronous induction ($M = 0.38$ cm, SD $= 2.25$), $t(123) = 2.18$, $p = 0.031$, 95% CI [0.1, 1.1], $d = 0.20$, $B_{\text{H}(0,1)} = 4.51$, RR $= [0.17,$ $1.68]$. Subjective report (S1-3) was higher for synchronous induction ($M = 0.9$, SD $= 1.6$) than asynchronous induction ($M$ $= -0.7$, SD $= 1.7$), $t(123) = 9.89$, $p < 0.001$, 95% CI [1.3, 1.9], $d$ $= 0.89$, $B_{\text{H}(0,1)} = 8.67 \times 10^{20}$, RR $= [0.02, 62]$. Therefore, according to a typical control procedure, the RHI was successfully induced in the control group.

Finally, we conducted pre-registered tests of relationships between RHI difference scores (synchronous induction minus asynchronous induction) and hypnotisability. There was only anecdotal evidence for the difference in proprioceptive drift score between induction conditions being related to hypnotisability in the control group, $b = 0.64$ cm drift/unit SWASH (SE $= 0.36$), $t$ $(122) = 1.80$, $p$ (one-tailed) $= 0.037$, 95% CI $[-0.06, 1.34]$, $R^2 = 0.026$, $B_{\text{H}(0,0.4)} = 2.91$, RR $= [0.4, 0.8]$. There was no sensitive evidence for or against a relationship between hypnotisability and the difference in mean subjective rating (S1–S3) between induction conditions in the control condition, $b = -0.002$ rating/unit SWASH (SE $= 0.23$), $t(122) = 0.011$, $p$ (one tailed) $= 0.496$, 95% CI $[-0.46, 0.46]$, $R^2 = 0.00$, $B_{\text{H}(0,0.3)} = 0.61$, RR $= [0, 0.75]$.

There was no evidence that the experimental manipulation caused expectancies to change in the predicted direction. Therefore, we were unable to manipulate participant expectancies as planned. We can only speculate as to why participants told to expect experience of the illusion for asynchronous induction did not believe the information they were given. Participants may already have had knowledge of the RHI, or there may be strong expectancies inherent in the task (e.g., synchrony of visual and tactile stimuli is common in experience of one's own hand). Additionally, our explanation of why participants should expect a greater effect when stimuli were asynchronous may have been unconvincing and simply not believed by participants. Consequently, we are unable to directly establish a causal role for phenomenological control in the RHI. However, we collected RHI and hypnotisability measures from 353 participants across groups and correlational evidence for substantial relationships between the RHI and hypnotisability in this unusually large sample would, considered alongside the correlational evidence reported here for mirror-sensory synesthesia, carry important implications for interpretations of the RHI. Furthermore, we have a measure of expectancies in the RHI illusion. Evidence that expectancies predict RHI measures would provide evidence which could not be easily explained by multimodal integration theories but which would be consistent with the theory that top-down processes are required, and that phenomenological control drives RHI

measures. As there was no evidence the expectancy manipulation worked, we henceforth analyse data collapsing across the manipulation (see Supplementary Table 1 for analyses of each condition which were entered into Bayesian meta-analysis to calculate statistics for collapsed conditions).

In our pre-registered analysis, we found no evidence for or against a relationship between an RHI agreement score difference measure (synchronous minus asynchronous induction). However, difference measures are rarely used in RHI research. Rather, asynchronous condition measures are typically used only in a prior check that suggestion and compliance effects have been controlled. That is, statistical evidence that synchronous condition measures are greater than asynchronous induction measures is used to justify the interpretation of synchronous induction measures as reflecting illusion response. We reviewed the twenty most highly cited RHI papers. 100% of subjective report measures and 65% of proprioceptive drift measures were reported in this way rather than using a measure of the difference between conditions as the measure of interest (see Supplementary Discussion for details). Therefore, our pre-registered measures were not well motivated by the literature. We therefore conduct simple correlational analyses of the measures most commonly reported in RHI research. That is, measures following synchronous induction (following a control test that asynchronous induction measures are lower than synchronous condition measures, as reported in our pre-registered analysis). While not pre-registered, these analyses are constrained by closely adhering to the most common procedures in RHI research and are necessary to support direct comparison with, and therefore inform interpretation of, the bulk of existing RHI literature.

Evidence that hypnotisability scores predict the most common RHI measures would be consistent with the theory that the RHI, as commonly reported, is at least partially driven by phenomenological control. Hypnotisability predicted synchronous induction subjective report of ownership and location (S1–S3; Fig. 3a), $b = 0.57$ rating/unit SWASH (SE $= 0.10$), $t(347) = 5.86$, $p < 0.001$, 95% CI [0.37, 0.77], $R^2 = 0.09$, $B_{\text{H}(0,1.4)} = 1.49 \times 10^6$, RR $= [0.02,$ $201.70]$ of visual hallucinations (S4; Fig. 3b), $b = 0.83$ rating/unit SWASH (SE $= 0.13$), $t(347) = 6.56$, $p < 0.001$, 95% CI [0.57, 1.09], $R^2 = 0.11$, $B_{\text{H}(0,1.4)} = 1.10 \times 10^8$, RR $= [0.023, 302.3]$ and of proprioceptive drift (Fig. 3c) $b = 0.58$ cm drift/unit SWASH (SE $= 0.21$), $t(347) = 2.65$, $p = 0.008$, 95% CI [0.17, 0.98], $R^2 = 0.02$, $B_{\text{H}(0,0.3)} = 14.66$, RR $= [0.10, 6.30]$.

Asynchronous induction measures are generally assumed to represent suggestion effects in RHI studies. If asynchronous measures are predicted by hypnotisability, this assumption would be supported. Hypnotisability also predicted asynchronous induction subjective report of ownership and location (S1–S3), $b = 0.70$ rating/unit SWASH (SE $= 0.11$), $t(347) = 5.86$, $p < 0.001$, 95% CI [0.48, 0.91], $R^2 = 0.09$, $B_{\text{H}(0,1.4)} = 4.30 \times 10^7$, RR $= [0.03,$ $251.89]$ and visual hallucinations, $b = 0.87$ rating/unit SWASH (SE $= 0.12$), $t(347) = 6.87$, $p < 0.001$, 95% CI [0.63, 1.11], $R^2 = 0.12$, $B_{\text{H}(0,1.4)} = 1.83 \times 10^{10}$, RR $= [0.03, 321.81]$. However, there was no evidence for or against a relationship between asynchronous induction proprioceptive drift and hypnotisability (Fig. 3c) $b = 0.02$ cm drift/unit SWASH (SE $= 0.25$), $t(347) = 1.03$, $p = 0.306$, 95% CI $[-0.48, 0.52]$, $R^2 = 0.003$, $B_{\text{H}(0,0.3)} = 0.67$, RR $= [0, 0.80]$.

Imaginative suggestion effects are driven by expectancies. Evidence that participant expectancies predict synchronous induction RHI measures would be consistent with the theory that the RHI as most commonly reported is attributable to phenomenological control. Expectancy rating ($-3$ to $+3$) for illusion experience in synchronous induction predicted subjective report for the synchronous induction (Fig. 3d), $b = 0.33$ rating/unit expectancy (SE $= 0.07$), $t(347) = 4.71$, $p < 0.001$, 95% CI

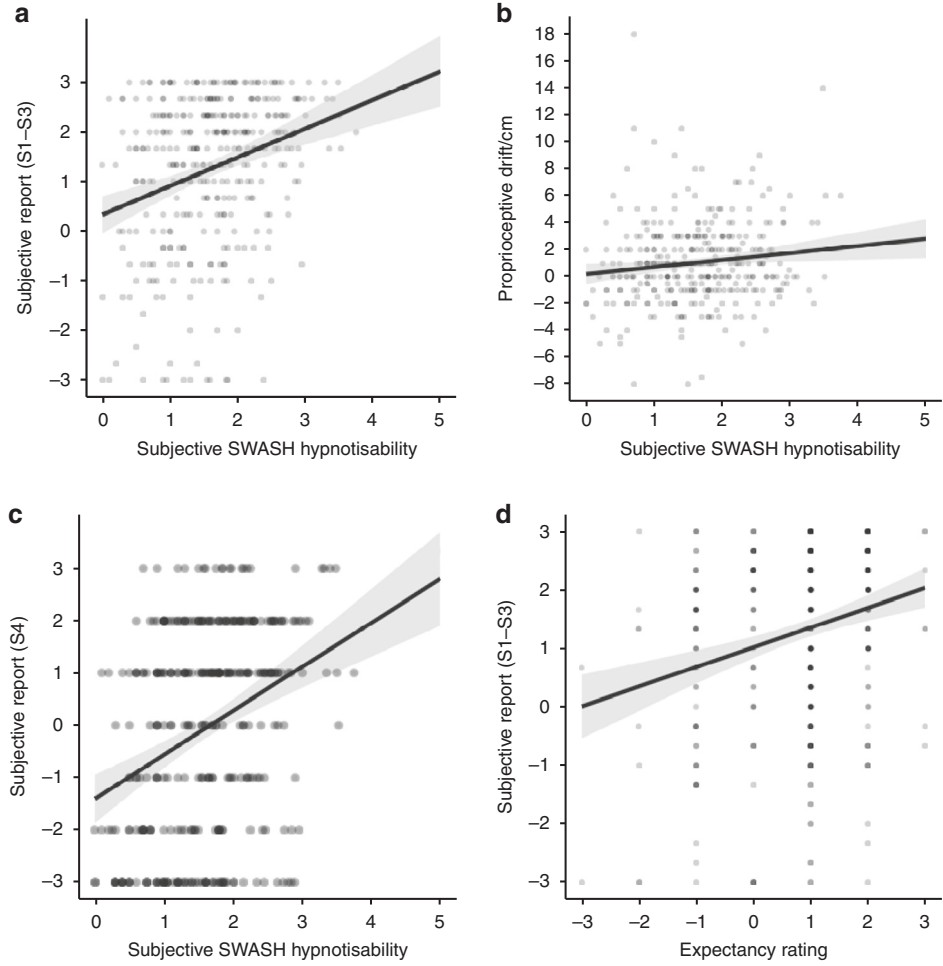

**Fig. 3 Scatter plots showing linear regression ($n = 353$ participants) of synchronous condition rubber hand illusion measures on hypnotisability and expectancies. a** Mean subjective report for location and ownership (S1–S3) on hypnotisability, $r_s = 0.26$, 95% CI [0.16, 0.37]. **b** Proprioceptive drift on hypnotisability, $r_s = 0.11$, 95% CI [0.003, 0.22], (**c**) subjective report for visual hallucination (S4) on hypnotisability, $r_s = 0.31$, 95% CI [0.21, 0.43], and (**d**) synchronous condition mean subjective report for location and ownership (S1–S3) on synchronous condition expectancy. $r_s = 0.23$, 95% CI [0.13, 0.34]. Error bars show 95% CI. The centre of credibility intervals is the predicted score. Note: SWASH is an abbreviation of Sussex-Waterloo Scale of Hypnotisability.

[0.20, 0.47], $R^2 = 0.06$, $B_{H(0,0.4)} = 1.66 \times 10^4$, RR = [0.017, 107.9]. There was no evidence for or against an effect of expectancy for the synchronous induction on synchronous induction proprioceptive drift, $b = 0.09$ rating/unit expectancy (SE = 0.14), $t(347) = 0.60$, $p = 0.551$, 95% CI = [−0.19, 0.37], $R^2 = 0.001$, $B_{H(0,0.65)} = 0.38$, RR = [0, 0.8].

## Discussion

We report evidence consistent with the theory that key measures in the field of embodiment are, at least partially, driven by phenomenological control. Measures of mirror-sensory synaesthesia and the rubber hand illusion were predicted by the ability to control phenomenology in a hypnotic context. The correlations were similar to those between individual hypnosis scale items and hypnotisability scales[37]. Therefore, while the evidence presented here is correlational, a simple explanation is that hypnotic responding and measures of these embodiment phenomena both reflect trait differences in the ability to control phenomenology according to expectancies. These embodiment effects may or may not be entirely attributable to demand characteristics and phenomenological control. Existing approaches to control for suggestion effects in these measures are not effective, and interpretation of RHI and mirror-sensory synaesthesia measures

requires consideration of trait differences in the ability to control experience to meet expectancies. In sum, because phenomenological control abilities are engaged within a range of contexts and in response to indirect or implicit suggestion, measures of experiential change in psychological studies may reflect top-down control of perception instead of, or in addition to other posited mechanisms.

Hypnotisability scores substantially predicted measures of mirror touch and pain. Further work will be required to identify the relative influence of expectancies, phenomenological control abilities and other mechanisms in mirror-sensory synaesthesia. For example, is mirror touch a less common experience than mirror pain because fewer people expect to experience it, because it is easier to generate experience of pain than of touch, or because of other mechanisms not related to phenomenological control? Mirror touch and pain experience also occurs outside the laboratory. Phenomenological control is sensitive to plans and goals[38,39], and it is a long-standing adage amongst hypnosis researchers that all hypnosis can be seen as self-hypnosis (e.g. Kilhstrom[40], pp 217–218). Therefore, people may implement phenomenological control in everyday life when it suits their plans and goals; namely when they have an interest in having a certain subjective experience; for example, in the case of vicarious pain the goal of being an empathic person. Notably, experiences

of touch and pain are common hypnotic suggestion effects: pain modulation is perhaps the most studied of all imaginative suggestion effects[41] and felt touch is a component of the most influential hypnotisability measurement scales (and their derivatives), in the form of a suggestion that an insect has landed upon the subject's hand[42,43].

Subjective report of ownership, location and visual hallucination following both synchronous and asynchronous induction were substantially predicted by hypnotisability. Consistent with evidence that RHI control measures are confounded by demand characteristics, Lush[36] found that the expectancies of naive participants only described and not given the RHI procedure matched the classic pattern of illusion experience (that is, illusion experience following synchronous but not asynchronous induction). That is, differences in report between illusion and control conditions may simply be attributable to differing expectancies and response difficulties of suggestion effects, and future RHI studies must ensure that expectancies and difficulty are closely matched across all control and illusion conditions to support claims that suggestion effects have been controlled. Existing evidence that expectation is sufficient to generate the RHI[44,45] (contrast Guterstam)[56] is parsimoniously explained if the RHI is an implicit imaginative suggestion effect.

Proprioceptive drift was predicted by hypnotisability for synchronous induction. Relationships between hypnotisability and synchronous induction measures were considerable for both subjective report and drift; for each 1 point increase in SWASH score (out of a maximum of 5) subjective report (S1–S3) increased by approximately half a point (7 point scale) and proprioceptive drift increased by approximately 0.5 cm. Two studies have explicitly related operational definitions of imaginative suggestion to the rubber hand illusion. Walsh et al.[46] reported a positive correlation between hypnotisability and proprioceptive drift. While noting that synchronous stimulation presented a strong implicit suggestion, they argued against a role for suggestion in drift on the basis of a non-significant correlation between subjective report and hypnotisability (though the direction of the reported effect is consistent with a relationship). Conversely, Marotta et al.[47] report a correlation between sensory suggestibility (a measure which has been shown to be related to hypnotisability[48,49]; contrast Tasso & Perez[50]) and the RHI. Here a non-significant correlation was used to argue that proprioceptive drift is not related to suggestion. Non-significant results do not constitute evidence for the null hypothesis[51] and these studies do not, therefore, provide evidence against relationships between RHI measures and hypnotisability.

These results are consistent with a cognitive account of the RHI, in which reports of illusory experience reflect acts of imagination[52] and with arguments that trait differences in the ability to respond to the rubber hand illusion are not attributable to multimodal sensory integration alone, but that top-down processes are required[35]. Walsh et al.[46] report a correlation of 0.55 between a drift difference measure and subjective hypnotisability. Consistent with evidence that proprioceptive drift is at least partly driven by top-down processes[33,53], Bayesian combination of our data and this prior result provides evidence that even when measured in this way, proprioceptive drift is related to hypnotisability ($r = 0.22$ (SE $= 0.08$). 95% CI [0.05, 0.38] $B_{H(0,0.3)} = 13.86$).

Previous findings can be reconstrued in terms of phenomenological control in the rubber hand illusion. For example, effects can occur in unimodal stimulation[54–56]; for mere expectations of touch;[44] or without a fake hand (e.g., the illusion can occur for drawings of hands[57], or empty space[58]). As in response to imaginative suggestions presented during hypnotisability screening, there are large individual differences in response to the illusion:

while some participants report powerful changes in experience, around 25% are apparently unable to experience the illusion at all[59,60]. Like response to imaginative suggestion, individual differences in response to the RHI reflect an underlying trait[35]. Here, we propose that this trait is the same as that measured by hypnotisability scales.

There is existing evidence that responses to mirror-sensory synaesthesia and response to the RHI are related. Mirror touch synaesthetes experience the rubber hand illusion[61] and the closely related enfacement illusion[62] in the absence of synchronous tactile signals. Similarly, mirror pain responders also report experiences of ownership over rubber hands following both synchronous and asynchronous stimulation[63] and show stronger effects in the rubber hand illusion than non-responders[64]. A simple explanation is that these effects each reflect underlying trait differences in the ability to control phenomenology to meet expectancies, as measured by hypnotisability scales.

If phenomenological control produced effects in embodiment experiments, the resulting reports of experiences would be genuine. There is plentiful evidence that response to imaginative suggestion within a hypnotic context involves genuine changes in experience, and this is not considered an open question by hypnosis researchers[65]. For example, the behaviour of high hypnotisables and lows attempting to simulate their behaviour is distinct[5,66], and hypnotic suggestion can result in apparently painless surgery[67]. However, response to imaginative suggestion is an interpretative and creative activity and subjective report measures may reflect a wide variety of experiences. For example, participants give varied responses to multiple choice questions about mirror-sensory synaesthesia[30], and the statement used to measure experiences of ownership over the rubber hand is open to a wide range of interpretations[68]. If reports reflect phenomenological control, the only constraint on experience is that it meets individual, interpretative expectancies, and quantitative measures of mirror-sensory synaesthesia and the rubber hand illusion are likely to conceal great variety in experience. To the degree to which measures of embodiment effects are driven by phenomenological control, they cannot provide a solid foundation for theories about embodiment (e.g., mirror neurons or the sense of ownership).

The evidence presented here is based on self-report measures, but there are established indirect measures of these effects. For example, the RHI has been reported to incur physiological effects such as changes skin conductance response[69] or histamine reactivity[70]. These measures are also likely to reflect phenomenological control; hypnotic suggestion can induce changes in skin conductance response[71] and histamine reactivity[72]. Brain imaging shows activity in areas considered to be important for body ownership experiences in the rubber hand illusion[73] and in mirror-sensory synaesthesia[30]. However, because imaginative suggestion also produces such changes (e.g., visual cortex activity for visual hallucinations[74] suggested pain shows similar patterns of activation to physically induced pain[64]), brain imaging data are also not evidence against suggestion effects.

While the effects reported here are related to embodiment and involve tactile hallucinations, there is no reason to expect that such relationships will be limited to such cases. An RHI control statement about changes in visual features of the hand (S4) was here predicted by hypnotisability and may be an example of demand characteristic driven subjective report compatible with a visual hallucination. Furthermore, the hypnotisability scale with which reports of mirror touch and pain correlate here contains ten suggestions, and only one of these (a suggestion that a mosquito can be felt on the participant's hand) involves tactile experience. Imaginative suggestion effects are not limited to somatosensory experience, and potentially any experience which

can be generated in response to imaginative suggestion in a hypnotic context could be generated to meet expectancies arising from demand characteristics (for a review of hypnotic suggestion effects, see Woody and Barnier[75]; for a review of top-down effects in imaginative recent suggestion, see Terhune et al.[4]).

While we report correlational evidence that trait hypnotisability (phenomenological control in the hypnotic context) predicts embodiment measures, we do not directly provide evidence for anything beyond these relationships. However, our inferences draw upon a vast literature investigating trait hypnotisability and response to imaginative suggestion and the proposal that these correlations indicate that response to the rubber hand illusion and mirror-sensory synaesthesia at least partially is a parsimonious theory (simple because it posits one mechanism to explain a wide range of phenomena) which cannot currently be ruled out. Experiments which measure behaviour linked to relatively high-level cognitive processes may be susceptible to the generation of experience in response to demand characteristics; for example, experimental paradigms in embodiment research closely related to the rubber hand illusion (e.g., the full body illusion). Similarly, given the classical suggestion effect of the experience of involuntariness, we should expect to find a relationship between reports of changes in the sense of agency[76] and phenomenological control. Indeed, mirror-touch synaesthetes are particularly vulnerable to manipulations of the experience of agency[77]. However, it cannot be ruled out that there may even be some phenomenological control effects in apparently low level tasks, if those tasks are not as free of top-down influence (for example, if demand characteristics have not been well controlled) as has been assumed[78,79]. These intriguing empirical questions remain open.

The concept of demand characteristics (unconscious changes in behaviour to meet implied experimental aims) as developed by Orne[80] is often used to refer to behavioural compliance effects specifically. However, Orne considered demand characteristics not to be limited to shifts in behaviour but to potentially include genuine changes in phenomenology in a substantial subset of the population. That experimental demand characteristics can act as implicit imaginative suggestions to generate changes in phenomenology has been overlooked. This study motivates a systematic exploration of the influence of phenomenological control across a wide range of experimental contexts in which subjective phenomenological reports are prominent.

The contextually driven control of phenomenology in accordance with expectations has been a topic of psychological study since the birth of the discipline. Unfortunately, this has been primarily restricted to the hypnotic context, with the result that this fundamental aspect of human psychology has not been seen as relevant to other areas of psychology. We argue that it is now necessary to look beyond the hypnotic context to explore phenomenological control in wider contexts, both to avoid accidentally engaging this ability when measuring other target phenomena and as an empirical target in its own right.

## Methods

**Participants**. For study 1, we combined databases of participants screened for vicarious pain on the Vicarious Pain Questionnaire or VPQ ($n = 1056$, age 18–60 years, $M = 20.42$, SD = 4.16, 759 females, 297 males; reported in Botan et al.[81]; Grice-Jackson et al.[28]) and for mirror touch synaesthesia or MTS ($n = 283$, age = 18–52 years, $M = 21.3$ years SD = 4.5, 237 females, 46 males; reported in Ward et al.[82]) with a database of participants screened for hypnotisability using the SWASH at Sussex University. For vicarious pain, 404 participant's scores were matched and for mirror touch synaesthesia, 154 were matched.

For study 2, an opportunity sample of 385 participants took part in a hypnosis screening procedure during an undergraduate Psychology laboratory practical session. 32 participants either declined to take part in a second rubber hand illusion session or were unable to upload hypnosis data due to a server error. Therefore, we recorded data for 353 participants across both procedures. Participants were assigned to one of three conditions according to which of three instructions would

be delivered by headphones. 114 participants (85 female, 29 male; mean age = 19.0, SD = 1.6) were informed that the effect would be strongest in synchronous stroking (synchrony instruction condition, though note this was not worded as a hypnotic suggestion but as information), 115 (92 female, 21 Male; mean age = 19.0, SD = 1.4) were informed that the effect would be strongest in asynchronous stroking (asynchrony instruction condition) and 115 received no instructions (control condition). Nine participants who received no instructions (either because they removed their headphones before delivery or due to computer error) were added to the control group, so there were 124 (99 female, 25 male, mean age = 18.9, SD = 1.1) participants analysed for the control group. These participants were retained because we stated in our pre-registration document (available from https://osf.io/xf6u4/) that we would use all available data. However, no statistical inferences differ between the control group with or without the additional participants. Participants were screened for hypnotisability on the computer SWASH[83]. Approval was received from the University of Sussex ethics committee and participants gave informed consent for the study.

**Materials**. For study 1, hypnotisability was measured using the Sussex-Waterloo scale of hypnotisability (SWASH)[37], a 10 item adaptation of the Waterloo-Stanford scale (WSGC)[84]. SWASH scores are measured on a dichotomous objective scale and a subjective scale of between 0 and 5 for each item[37].

For full details of the procedure used to measure vicarious or mirror-pain, see Grice-Jackson et al.[28]. Participants watched 16 short video clips of people experiencing physical pain (e.g. falls, sports injuries, injections). After each video, participants responded to questions by computer. First they were asked if they experienced a bodily sensation of pain while viewing the video. If they answered yes, they were then asked to describe their pain by answering three further questions: (1) how intense their pain experience was (on a Likert scale from 1–10); (2) if pain was either "localised to the same point as the observed pain in the video", "localised but not to the same point", or "a general/non-localisable experience of pain"; (3) to select pain adjectives from a list that best described their vicarious pain experience (10 sensory descriptors, 10 affective descriptors and 3 cognitive-evaluative descriptors).

For full details of the procedure used to measure mirror touch synaesthesia, see Ward et al.[82]. Participants watched short video clips[85], including depictions of touch to a human (14 videos), touch to inanimate objects (dummies and a fan; 6 videos) and painful stimuli (e.g., injections; 6 videos). After each video, participants responded to questions by computer. First they were asked if they experienced anything on their body (excluding feelings of unease, disgust, or flinching). If they answered yes, they were asked three further questions: (1) Whether they would describe the sensation as Touch (without pain); Pain (without touch); Painful touch; Tingling; Itchiness; Feeling of being scratched; or Other. (2) Where on their body the sensation was felt (from a list of options). Finally, they were asked how intense the sensation was (on a Likert scale from 1–10).

For study 2, participants were screened for hypnotisability by computer. SWASH scores are numerically similar to in-person delivery whether delivered in a lab setting (PL, RBS, ZD in preparation) or delivered online[83]. SWASH computer induction is identical to the live delivery described in Lush et al.[37] except participants receive a recorded induction via headphones and responses are recorded using a computer keyboard rather than a paper form. Two changes are made to the stimuli and report procedure; a slide of 3 coloured balls is displayed on a computer screen rather than a lecture hall screen and the post-hypnotic suggestion changed from drawing a tree to pressing the keyboard space bar six times.

Printed materials (e.g., response sheets) and instruction scripts for study 2 are available at https://osf.io/huwxd/. The setup was based on experimental materials described in Botan et al.[63]. Participants sat at a table opposite the experimenter and placed their right hand in a black-fabric covered box ($61 \times 46 \times 23$ cm³). A realistic model hand was placed in the box so that it was in front of the participant's torso. Participants were asked to rest their right hand in a similar position to the rubber hand, with their index finger on a white dot positioned 20 cm lateral to the index finger of the fake hand. Participants were unable to see their own hand inside the box and could only view the fake hand through a square window in the top surface of the box.

There were five copies of the experimental setup, which were used simultaneously. In total, sixteen experimenters performed the experiment across eleven one hour sessions. Experimenters were blind to the instruction condition for each participant.

Before the experiment began, participants received pre-recorded information about the experiment via headphones. All participants were told that told that they were going to take part in a rubber hand illusion study and that this was a separate study to the hypnotisability screening procedure. Participants were pseudo-randomly assigned by a computer program to one of three groups. Participants in instruction groups were told that the illusion would be strongest in either the synchronous or asynchronous induction. Participants in the synchronous condition were told that illusion would be strongest when the felt and seen brush strokes occurred simultaneously because the relative timing of visual and tactile information influences whether or not the information is perceived as arising from a common source. Participants in the asynchronous condition were told that the illusion would be strongest when the felt brush stroke followed the seen brush

**Table 1 Statements, questions and response labels used to generate subjective report scores for expectancies and illusion experience.**

| Statement or question | Response labels |
|---|---|
| E1. How strongly do you expect to feel the rubber hand is your own hand at least a little bit when the brush strokes on your own hand and on the rubber hand **are** in synchrony? | 3 I am certain I will feel some effect<br>2 I am fairly certain I will feel some effect<br>1 I think I will feel some effect |
| E2. How strongly do you expect to feel the rubber hand is your own hand at least a little bit when the brush strokes on your own hand and on the rubber hand **are not** in synchrony? | 0 I have no idea either way<br>−1 I think I won't feel any effect<br>−2 I am fairly certain I won't feel any effect<br>−3 I am certain I won't feel any effect |
| S1. It seemed as if I were feeling the touch of the paintbrush in the location where I saw the rubber hand touched | 3 Strongly agree<br>−3 Strongly disagree |
| S2. It seemed as though the touch I felt was caused by the paintbrush touching the rubber hand | |
| S3. I felt as if the rubber hand were my hand | |
| S4 (control)The rubber hand began to resemble my own (real) hand, in terms of shape, skin tone, freckles or some other visual feature | |

stroke because of neural delays involved in integrating visual and tactile stimuli and that "researchers have found a 1/3 of a second to be the optimal delay, which we will approximately reproduce here in order to generate the maximum effect". The control group received no further information. See Supplemental Method for instruction condition scripts.

Participants recorded their participant number and their response to two expectancy questions on a response sheet (Table 1). Response sheets were folded after each response was recorded to prevent participants seeing their previous responses. The experimenter then recorded which of the two conditions (synchronous or asynchronous induction) was performed first (this order was alternated by participant). Pre-test and post-test proprioceptive drift were recorded immediately before and immediately following induction. First, the viewing window was then covered and a 60 cm ruler placed on the top of the box, with the numbers facing the experimenter. To prevent participants repeating responses for each measurement, the ruler was positioned so that the end was over the edge of the box by a variable distance, which was subtracted from the participant's report. Participants were told that the ruler would change position each time and that they should not try to report the same number repeatedly as it would not refer to the same position. They were then asked to point to the position at which they thought the index finger of their right hand was located and this was recorded to the nearest half cm. The cover was then removed and the participant was told to keep their gaze on the rubber hand and to focus on it. A paintbrush was then used to stroke the middle finger of the rubber hand at approximately 1 Hz while simultaneously stroking the middle finger of the participant's real hand with an identical brush either synchronously or with an asynchrony of approximately 0.5 Hz. Brush strokes were applied from the knuckle to the finger nail of the real and fake hands. This induction procedure continued for 60 s, following which participants were asked to keep their hand still and inside the box and the hole was covered. Participants were again asked to report the position of their real hand and then reported their level of agreement with statements S1–S4 on the response sheet (Table 1). The procedure was then repeated in the other condition (synchronous or asynchronous induction).

**Measures**. For study 1 we report analyses of the primary VPQ measure used to identify vicarious pain perceivers[28]: total pain response (the number of videos from 0 to 16 for which a bodily sensation of pain was reported and also the intensity of pain). For Mirror touch synaesthesia we report the mean number of reports of experience when observing touch to a human (0–14) and the mean number of responses when observing touch to an inanimate object (0–6).

For pre-registration of planned measures and planned analyses for study 2, see: https://osf.io/xf6u4/. Participants recorded responses on a 7 point Likert scale (Table 1). Subjective response for each participant was calculated from the average of S1–S3 and, separately, the score for S4. Proprioceptive drift was calculated by subtracting post induction stroking from pre induction stroking for each induction condition. Difference scores were calculated by subtracting synchronous condition measures from asynchronous condition measures. Subjective scale SWASH scores were used as a measure of hypnotisability due to greater reliability than the objective scale[37].

For whole sample exploration, participants from all three groups were combined in a Bayesian fixed-effects meta-analysis. Exploratory analyses were conducted on subjective report and proprioceptive drift following synchronous induction.

**Analyses**. For study 1, Bayes factors for raw slopes of VPQ and MTS measures regressed onto subjective SWASH score were modelled on the expected change in y (given by mean score on the VPQ measure across the whole sample) and the expected difference in SWASH score between 0 and the 50th percentile of the

sample (a SWASH score of 1.7) as the change in x. The maximum slope was therefore calculated in each case using the mean to bottom of scale for Y over the distance from mean to bottom of scale for X (the ratio-of-means heuristic described in Dienes)[86]. For VPQ measures, H1 was therefore modelled using half the maximum slopes of 3.9 over 1.7 (1.1) for total pain response and 1.0 over 1.7 for pain intensity (or 0.3) For MTS measures, H1 was modelled using half the maximum slopes of 1.5 over 1.7 (or 0.5) touch to a human and a maximum slope of 0.2 over 1.7 (or 0.07) for touch to an inanimate object.

To indicate the robustness of Bayesian conclusions, for each *B*, a robustness region is reported, giving the range of scales that qualitatively support a given conclusion (i.e. evidence as insensitive, or as supporting H0, or as supporting H1), notated as: RR [x1, x2] where x1 is the smallest SD that supports the conclusion and x2 is the largest.

Pre-registered analyses for Study 2 were not performed that were dependent upon the results of manipulation checks (outcome neutral tests) that in fact did not provide evidence that the manipulation worked. A first manipulation check tested the prediction that, given no explicit instructions, any difference in dependent variables in the control group would be due to a difference in expectations. For this, we tested the interaction of instruction condition (synchronous or asynchronous instructions) by rubber hand induction type (synchronous vs asynchronous) on expectancies. A Bayes factor was calculated using the difference in expectancy ratings between conditions (synchronous—asynchronous) in the control group to model the SD of a half-normal distribution with a mean of 0.

Further manipulation checks tested the prediction that in the control group, rubber hand illusion measures should be greater for synchronous than asynchronous induction. Bayes factors of *t*-tests for these differences were calculated using a model of H1 based on the reported difference in Botan et al.[63], across all groups (weighted by sample size); for proprioceptive drift, a difference of approximately 10 mm between synchronous and asynchronous condition and for subjective report, a difference in agreement score) of approximately 1 unit (7 point scale).

For Study 2 pre-registered key tests, we tested the prediction that the control group difference in illusion measures between synchronous and asynchronous inductions would be positively related to hypnotisability scores by regressing the effect of induction condition (synchronous—asynchronous) on illusion measures on SWASH hypnotisability rating. Raw regression slopes were tested by Bayes factors with H1s modelled on the effect size for the rubber hand illusion reported in Botan et al.[63] as the expected change in y and the expected difference in SWASH score between 0 and the 50th percentile as the expected change in x (cf. ratio-of-means heuristic recommended by Dienes)[86]. However, for consistency with the rest of the manuscript, here we calculated Bayes factors using this expected slope as a plausible maximum. In each case the predicted value for the pre-registered analysis was contained within the robustness region for the presented analysis, so this change makes no difference to inferences. The maximum slope for proprioceptive drift was therefore 13 mm over 1.6 (the difference in SWASH score between the 0 and the 50th percentile), or approximately 8 mm for each 1 point increase (out of 5) in SWASH subjective score, so H1 was modelled with an SD of 4 mm. The maximum slope for subjective report was calculated as 1 over 1.6, or 0.6 units per 1 point increase in SWASH subjective score, so H1 was modelled with a mean of 0 and an SD of 0.3. These a priori estimates appear plausible according to the correlation between proprioceptive drift and hypnotisability reported in Walsh et al.[46] and the correlation between subjective report and sensory suggestibility reported in Marotta et al.[47].

For study 2 exploratory analyses, first we report regression analyses of subjective report and proprioceptive drift scores on hypnotisability to test the hypothesis that the ability to respond to suggestion in a hypnotic context is positively related to RHI measures. Next, we report regression analyses of subjective report and proprioceptive drift for synchronous induction on expectancies for synchronous

induction to test the prediction that, if the RHI is a phenomenological control effect, expectancies will positively predict illusion response. Here we present meta-analytically combined results for synchronous induction only.

Bayes factor robustness regions for Study 2 exploratory analyses are reported as in Study 1. Raw regression slopes were tested by Bayes factors with H1s modelled as half-normal distributions on the expected change in y (given by mean score across the whole sample) and the expected difference in SWASH score between 0 and the 50th percentile (a SWASH score of 1.6) as the expected change in x. A Bayes factor for regression of subjective report on subjective SWASH score was calculated using half the maximum slope of 4.3 (mean synchronous induction subjective report) over 1.6, or 1.4. Similarly, a Bayes factor for regression of proprioceptive drift on subjective SWASH score was calculated using half the maximum slope of 1 (mean synchronous induction drift) over 1.6, or 0.3. A Bayes factor for regressions of synchronous induction subjective report or proprioceptive drift on expectancy rating for synchronous induction was calculated using half the maximum slope given by the distance from the mean to bottom of scale for Y over the distance from the mean to the bottom of scale for X: 3.8 over 4.3, or 0.4 for subjective report and 3.8 over 3, or 0.65 for proprioceptive drift. A Bayes factor for this analysis was therefore calculated using half this value as the SD of a half-normal, or 0.65. All other analyses are based on the interpretation of 95% CIs. For whole sample analyses these are meta-analytically combined. In each case, CIs are interpreted as 95% credible intervals with a uniform prior. All analyses except Bayes factors were calculated using JASP 0.9.2 (ref. [87]). Bayes factors were calculated using Dienes' Bayes factor calculator at http://www.lifesci.sussex.ac.uk/home/Zoltan_Dienes/inference/Bayes.htm.

**Reporting summary**. Further information on research design is available in the Nature Research Reporting Summary linked to this article.

## Data availability statement

The data that support the findings of this study are available at https://osf.io/huwxd/. All figures have associated raw data. There are no restrictions on data availability.

## Code availability statement

The hypnotisability screening software and installation instructions are available at https://osf.io/huwxd/.

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

## Acknowledgements

Zoltan Dienes and Peter Lush were funded by grant 163/18 from the BIAL foundation. Peter Lush and Anil Seth were supported by the Dr Mortimer and Theresa Sackler Foundation and the Canadian Institute for Advanced Research (CIFAR) Azrieli Programme on Brain, Mind, and Consciousness. Thanks to Warrick Roseboom for predicting a relationship between mirror-sensory synaesthesia and hypnotisability. Thanks to Robert Avery, Reny Baykova, Phoebe Bazzard, Anthony Collins, Giulia Esposito, Phillip Kaniuth, Giuseppe Lai, Mengze Li, Isabel Maranhao, Julie McDermott, Christopher Osborne, Gabriel Rollison, Konstantinos Seintaridis, Maxine Sherman and Nicholas Young for data collection.

## Author contributions

P.L. and Z.D. developed the study concept. P.L., R.B.S., V.B. and Z.D. designed Study 2. P.L. and R.B.S. collected experimental data for Study 2. P.L. conducted analyses and drafted the manuscript, and Z.D., A.K.S., R.B.S., V.B. and J.W. provided critical revisions.

## Competing interests

The authors declare no competing interests.
