## [Peer Review File · Nature Communications]

Reviewers' Comments:

Reviewer #1:

Remarks to the Author:

This study provides very interesting data on the role of hypnotizability and expectancy in the mirror-sensory synesthesia and the rubber hand illusion. The presentation is overlong and is marred by some statistical problems, but these could be remedied in a revision.

The major statistical problem is the conversion of continuous scores into discrete categories for the analyses (i.e., converting hypnotizability scores into quartiles and dichotomizing synesthesia scores). Statisticians who have written on this subject (beginning with Jacob Cohen in 1983) have been unanimous in arguing against this practice. It decreases power substantially (equivalent to discarding 1/3 or more of the data), and in some cases can also lead to false positives. Most of these articles are directed to dichotomization of continuous variables, but Streiner (*Can J Psychiatry* 2002;47:262–266) has discussed this also for categorizing them into more groups. These analyses should be redone using the raw scores rather than quartiles.

The authors claim that the data were obtained "from over 1000 participants, and our analyses are therefore based on data from samples much larger than are typical in laboratory-based experimental psychology tasks." Reading the rest of the text, it becomes clear that the analyses were performed on datasets of less than half that size (e.g., $n=404$ for vicarious pain and 353 for the rubber hand illusion). Therefore, that claim should be removed.

The authors "can only speculate as to why participants told to expect experience of the illusion for asynchronous induction did not believe the information they were given." It would be helpful if the authors could provide more information about the information given to participants to induce this expectation. Did they provide a convincing rationale? It seems to me that synchronous touch would be more believable without any attempt to provide a contrary rationale.

Finally, I found the many parts of the manuscript difficult to decipher. Some careful editing would be helpful.

Reviewed by Irving Kirsch (signed review)

Reviewer #2:

Remarks to the Author:

This paper reports that individuals who are high in hypnotisability are more subject to expectancy effects in two somatosensory illusions. It reflects a heroic effort by the researchers (e.g., in terms of sample size), and the results will certainly be of interest to scholars interested in those illusions. It is also admirable that the researchers pre-registered the studies.

But it was much harder for me to get on board with the broader conclusions the authors advertise; the rhetoric and conclusions just seem to go much too far beyond what is justified by the data. One cannot conclude from the fact that hypnotisable people show greater expectancy effects in the rubber hand illusion that one's work will "transform human behavioural science" (!), a conclusion that is repeated throughout the paper in various forms. This was a problem for me throughout the paper, and so I think do think it is better suited for a more specialized journal than *Nature Communications*. My

comments below are mostly about that worry, rather than the studies themselves, which seem beautifully and competently executed (and I certainly commend the authors on that). The issue is about the interpretation.

- My worry about the authors' conclusions begins even with the title of the paper, whose theme is repeated throughout: I don't see how this paper can claim to have demonstrated "phenomenological control" in the first place. "Control" is just not what is under study here -- what is being discussed are expectancy effects, which may be completely beyond the control of the subject. Hypnosis itself might well be a case of "control"; but that's not what the authors are investigating here -- the work concerns the interaction between hypnotisability and expectation. I don't know if this appears to be nitpicky, but I hope not; my comments here concern the core, fundamental claim of the paper. Control is different than expectancy effects.

-- This matters because readers who learn about, but do not carefully study, this paper, will be apt to completely misinterpret it for exactly this reason. The authors hope that other researchers will account for the present findings in running their own studies; but these claims about "control" will surely mislead those other researchers, just as it misled me before I actually read the paper's details. It will make readers/researchers think that subjects in their experiments can "control" their phenomenology, and that this must be accounted for. But nothing like that has been demonstrated, so confusion can only result from this.

-- Moreover, this is not easy to fix: The problem I'm suggesting exists here would require completely reframing the paper from the ground up (though of course the experiments and results could remain the same). So it's not a matter of, e.g., just changing the title; if my worry is justified, the whole framing and rhetoric would have to change, in ways that would just make this a new paper altogether.

- This kind of worry extends even to the more substantive claims in the paper. One limitation of the study that the authors seem not to properly explore and acknowledge is that the two illusions under study are both somatosensory/tactile illusions, both of which require very unusual experimental circumstances. Again, the authors' ambition for this project is, as written, to "transform human behavioral science"; with consequences like those hanging in the balance, it seems crucial to demonstrate some kind of generalization from these very unusual and isolated conditions to other sensory modalities (such as vision and audition), other tasks, and also just far more normal circumstances than the craftiness required for the present illusions. Human behavioral science (= psychology, roughly?) is so incredibly broad; a relationship between hypnotisability and some (admittedly bizarre) tactile illusions just doesn't engage with those stakes.

-- A telling example of this, for me, appears on p.25: "these cases are intended as examples to draw attention to possible roles of phenomenological control effects across behavioural science. Any study in which the experimenter's expectations are discernible to the participant and for which the generation of an experience could, in principle, influence measures is likely to be affected by trait differences in phenomenological control". But these cases seem so unlikely to succeed as "examples" of this, because they are unusual and constrained -- they just don't look much like a normal everyday study in psychology. How do the researchers know that "any study" with discernible expectations is "likely to be affected" by phenomenological control? Nothing in the present work establishes or even suggests this. There's no evidence that this applies beyond illusions, somatosensory phenomena, etc. "Drawing attention" is one thing; this paper does indeed *raise this possibility*. But it does so very weakly, and with no basis to claim what is "likely" or unlikely about the rest of psychology.

-- The authors do acknowledge that it will be important to test "low-level tasks", and they write "We do not know the extent of phenomenological control effects". But then they follow this up immediately with the same kind of rhetoric: "they cannot be ruled out in any study involving human report"; indeed, that's true, but that's not the same as suggesting that they are likely to appear in those studies! That requires (much) more evidence.

- So far, I have commented on the fact that the researchers' claims seem to go too far beyond what is given in the data. Indeed, one reason to suspect that these results may not generalize is that other sensory modalities tend to show much less susceptibility to the sorts of illusory phenomena explored here. There is a very active debate, for example, about whether vision and audition are subject to top-down phenomenological control in the first place (e.g., Dunning & Balci, 2013, *Current Directions in Psychological Science*; Firestone & Scholl, 2016, *Behavioral and Brain Sciences*; Lupyan, 2015, *Review of Philosophy and Psychology*; MacPherson, 2012, *Philosophy and Phenomenological Research*; Norris, McQueen, & Cutler, 2000, *Behavioral and Brain Sciences*; Proffitt, 2006, *Perspectives on Psychological Science*), with many arguing that such effects do not and cannot occur, and others arguing that they always or often occur. I do not know of similar arguments in the tactile domain, because it's my sense that researchers tend to think that somatosensory phenomena are more subject to top-down influence. But if that's the case, then these results may not generalize as far as the authors imagine. A future version of this paper would make a more powerful argument about perception as a whole if it were better engaged with the broader literature on top-down effects on perception.

I'm sorry that this review has been mostly negative; I do have a positive impression of the work. But for me the conclusions the authors attach to it are just far too strong and broad to justify publication in this form.

Reviewer #1 (Remarks to the Author):

This study provides very interesting data on the role of hypnotizability and expectancy in the mirror-sensory synesthesia and the rubber hand illusion. The presentation is overlong and is marred by some statistical problems, but these could be remedied in a revision.

We thank the reviewer for their favourable evaluation. The manuscript has been revised, and clarification about statistical analyses added (see below).

The major statistical problem is the conversion of continuous scores into discrete categories for the analyses (i.e., converting hypnotizability scores into quartiles and dichotomizing synesthesia scores). Statisticians who have written on this subject (beginning with Jacob Cohen in 1983) have been unanimous in arguing against this practice. It decreases power substantially (equivalent to discarding 1/3 or more of the data), and in some cases can also lead to false positives. Most of these articles are directed to dichotomization of continuous variables, but Streiner (Can J Psychiatry 2002;47:262–266) has discussed this also for categorizing them into more groups. These analyses should be redone using the raw scores rather than quartiles.

The reviewer is mistaken about the statistical analyses conducted. Figure 4b and figure 4c show data in split groups, but the data are split by quartiles merely for illustrative purposes. As the reviewer recommends, our inferences are all based on analyses of raw scores (correlation and linear regression). We have added a note to the figure 4 legend:

“Please note that the data are here split by quartile for illustrative purposes. No statistical analyses have been conducted with groups split by quartiles.

The authors claim that the data were obtained “from over 1000 participants, and our analyses are therefore based on data from samples much larger than are typical in laboratory-based experimental psychology tasks.” Reading the rest of the text, it becomes clear that the analyses were performed on datasets of less than half that size (e.g., n=404 for vicarious pain and 353 for the rubber hand illusion). Therefore, that claim should be removed.

The reviewer is quite right that this text is potentially misleading. This was not our intention and we apologise for this oversight. The given number was for the combined samples across the three samples (vicarious pain, mirror-touch synaesthesia and the rubber hand illusion).

Abstract text changed from:

“two large sample studies (total 1039)”

to

“In large sample studies (of 156, 404 and 353 participants)”

Deleted from p.8:

~~This approach has been taken in each case; across all three studies we collected data from over 1000 participants, and our analyses are therefore based on data from samples much larger than are typical in laboratory based experimental psychology tasks.~~

The authors “can only speculate as to why participants told to expect experience of the illusion for asynchronous induction did not believe the information they were given.” It would be helpful if the authors could provide more information about the information given to participants to induce this expectation. Did they provide a convincing rationale? It seems to me that synchronous touch would be more believable without any attempt to provide a contrary rationale.

We thank the reviewer for drawing our attention to this omission. The text of the expectancy manipulation has been added to the supplemental material. The following text has been added to the Methods (p.33):

“Participants in the synchronous condition were told that illusion would be strongest when the felt and seen brush strokes occurred simultaneously because the relative timing of visual and tactile information influences whether or not the information is perceived as arising from a common source. Participants in the asynchronous condition were told that the illusion would be strongest when the felt brush stroke followed the seen brush stroke because of neural delays involved in integrating visual and tactile stimuli and that “researchers have found a 1/3 of a second to be the optimal delay, which we will approximately reproduce here in order to generate the maximum effect”.

Added to p.15:

“Additionally, our explanation of why participants should expect a greater effect when stimuli were asynchronous may have been unconvincing and simply not believed by participants”

Finally, I found the many parts of the manuscript difficult to decipher. Some careful editing would be helpful.

We have edited the manuscript to improve clarity.

Reviewer #2 (Remarks to the Author):

1. This paper reports that individuals who are high in hypnotisability are more subject to expectancy effects in two somatosensory illusions. It reflects a heroic effort by the researchers (e.g., in terms of sample size), and the results will certainly be of interest to scholars interested in those illusions. It is also admirable that the researchers pre-registered the studies.

We thank the reviewer for this positive evaluation.

2. But it was much harder for me to get on board with the broader conclusions the authors advertise; the rhetoric and conclusions just seem to go much too far beyond what is justified by the data. One cannot conclude from the fact that hypnotisable people show greater expectancy effects in the rubber hand illusion that one's work will "transform human behavioural science" (!), a conclusion that is repeated throughout the paper in various forms. This was a problem for me throughout the paper, and so I think do think it is better suited for a more specialized journal than Nature Communications. My comments below are mostly about that worry, rather than the studies themselves, which seem beautifully and competently executed (and I certainly commend the authors on that). The issue is about the interpretation.

We thank the reviewer for their generous appraisal of the studies. Their concerns are understandable and are due to an unfortunate framing for which we take full responsibility. We fully agree that more cautious language is required at this stage and have removed the offending claim (see below). However, we do not agree that this makes the paper merely of specialist interest. Even if only the reported effects are considered, the results prompt the reappraisal of extensive literatures on embodiment. They would also need to be taken into account in the design of future experiments testing changes in experience. The reviewer is correct that we do not have evidence that the issue extends beyond these cases, but the possibility that it may certainly can not be ruled out from the armchair. If the relationships seen here do extend to other cases the potential implications are considerable, but we agree that this possibility should be handled much more carefully in the manuscript.

- My worry about the authors' conclusions begins even with the title of the paper, whose theme is repeated throughout: I don't see how this paper can claim to have demonstrated "phenomenological control" in the first place. "Control" is just not what is under study here -- what is being discussed are expectancy effects, which may be completely beyond the control of the subject. Hypnosis itself might well be a case of "control"; but that's not what the authors are investigating here -- the work concerns the interaction between hypnotisability and expectation. I don't know if this appears to be nitpicky, but I hope not; my comments here concern the core, fundamental claim of the paper. Control is different than expectancy effects.

-- This matters because readers who learn about, but do not carefully study, this paper, will be apt to completely misinterpret it for exactly this reason. The authors hope that other researchers will account for the present findings in running their own studies; but these claims about "control" will surely mislead those other researchers, just as it misled me before I actually read the paper's details. It will make readers/researchers think that subjects in their experiments can "control" their phenomenology, and that this must be accounted for. But nothing like that has been demonstrated, so confusion can only result from this.

-- Moreover, this is not easy to fix: The problem I'm suggesting exists here would require completely reframing the paper from the ground up (though of course the experiments and results could remain the same). So it's not a matter of, e.g., just changing the title; if my worry is justified, the whole framing and rhetoric would have to change, in ways that would just make this a new paper altogether.

Fortunately, the reviewer's concern is not justified, but arises from a misunderstanding about the status of the claim that response to imaginative suggestion involves control of phenomenology. We have failed to communicate the meaning of the term 'phenomenological control' to researchers unfamiliar with the imaginative suggestion literature, but this issue is easily rectified. This term was chosen to reflect current scientific consensus about imaginative suggestion effects. There is a large body of evidence that imaginative suggestions are under voluntary control and that hypnotisability scores reflect voluntary control of experience. This is not a controversial claim in imaginative suggestion research; we employ the term phenomenological control in an invited review paper for a special issue of *Psychology of Consciousness* summarising the current state of hypnosis research (Phenomenological control as cold control, Dienes et al, in press). No hypnosis researchers have yet taken issue with this terminology. Rather than requiring a substantial revision of the manuscript, this point can be addressed by the addition of justification of the use of the word 'control' to describe imaginative suggestion effects.

Added to introduction (p.3-4)

“In sum, response to imaginative suggestion involves the top-down voluntary control of action and experience (which is experienced as involuntary) according to a stable trait ability, and ‘hypnosis’ procedures are not necessary for successful responding (see Halligan and Oakley 2014 for a review of the evidence that hypnosis is just one context within which imaginative suggestion effects occur). Note that there is consensus among hypnosis researchers that response to imaginative suggestion involves top-down control. For example, in the Oxford Handbook of Hypnosis (2008), all five chapters in the section on "theoretical perspectives" involve control as a key part of the theories described. So our position is not a minority one; it is just accepting the current predominance of evidence and approach in the field of imaginative and hypnotic suggestion. Because the term ‘imaginative suggestibility’ evokes unwarranted association with other forms of suggestibility, we here refer to context-general top down effects on perception in response to imaginative suggestion as ‘phenomenological control’. We appreciate that readers unfamiliar with the hypnosis and imaginative suggestion literature are likely to have questions about the relationships between control processes, imaginative suggestion and apparently involuntary experience. Please see Dienes et al (in press) for detailed discussion of the concept of phenomenological control in response to imaginative suggestion.”

Added to p.5

“There is existing evidence that demand characteristics can drive experience in scientific experiments. For example, psychedelic experiences following placebo hallucinogen administration (Heaton, 1975; Olson et al, 2020) and hallucinated taste of salt in a psychophysical task Juhasz & Sarbin, 1966). It has previously been

proposed that such effects may be driven by the same mechanisms as response to imaginative suggestion (see Kirsch & Council, 1989; Michael, Garry, and Kirsch (2012). However, although this proposal is consistent with many theoretical accounts of hypnosis and has potentially wide-ranging implications (see Dienes et al, in press), it has not yet been directly investigated.

Also added to p.3:

“It has long been known that response to imaginative suggestion does not require hypnosis (Hull, 1933).”

3. - This kind of worry extends even to the more substantive claims in the paper. One limitation of the study that the authors seem not to properly explore and acknowledge is that the two illusions under study are both somatosensory/tactile illusions, both of which require very unusual experimental circumstances. Again, the authors ambition for this project is, as written, to "transform human behavioral science"; with consequences like those hanging in the balance, it seems crucial to demonstrate some kind of generalization from these very unusual and isolated conditions to other sensory modalities (such as vision and audition), other tasks, and also just far more normal circumstances than the craftiness required for the present illusions. Human behavioral science (= psychology, roughly?) is so incredibly broad; a relationship between hypnotisability and some (admittedly bizarre) tactile illusions just doesn't engage with those stakes.

*-- A telling example of this, for me, appears on p.25: "these cases are intended as examples to draw attention to possible roles of phenomenological control effects across behavioural science. Any study in which the experimenter's expectations are discernible to the participant and for which the generation of an experience could, in principle, influence measures is likely to be affected by trait differences in phenomenological control". But these cases seem so unlikely to succeed as "examples" of this, because they are unusual and constrained -- they just don't look much like a normal everyday study in psychology. How do the researchers know that "any study" with discernible expectations is "likely to be affected" by phenomenological control? Nothing in the present work establishes or even suggests this. There's no evidence that this applies beyond illusions, somatosensory phenomena, etc. "Drawing attention" is one thing; this paper does indeed *raise this possibility*. But it does so very weakly, and with no basis to claim what is "likely" or unlikely about the rest of psychology.*

-- The authors do acknowledge that it will be important to test "low-level tasks", and they write "We do not know the extent of phenomenological control effects". But then they follow this up immediately with the same kind of rhetoric: "they cannot be ruled out in any study involving human report"; indeed, that's true, but that's not the same as suggesting that they are likely to appear in those studies! That requires (much) more evidence.

Reading the reviewer's comments, we can see that our comments are unintentionally hubristic. What was intended as additional speculative consideration of possible outcomes of future research has been taken to be a primary claim. We take full responsibility for this misunderstanding, and feel we have failed to communicate our intended claim. We do not claim that we have provided evidence that phenomenological control effects confound all psychological experiments, or that the present study will by itself "transform behavioural science". Our claim is that the most parsimonious explanation for the relationships between

hypnotisability and embodiment measures is that a context-general ability to control phenomenology to meet expectancies. We do expect to find similar effects in other comparable 'illusions' (whenever people are asked to report unusual experiences). Additionally, it is also possible that phenomenological control may occur in a wide variety of psychological paradigms, but further research will be required to establish whether this is the case. We emphasise that our discussion of wider implications, which, as R2 rightly points out are only weakly supported by the evidence presented in the paper, are only peripheral claims. While these concerns of R2 are important and well-made, they can be easily rectified through altering the abstract, some lines in the introduction and the final paragraphs of the discussion. The substance of the manuscript need not change. The following changes have been made:

Removed from the abstract:

~~“and is likely to confound measures across behavioural science.”~~

We have removed the following lines from the manuscript:

Removed from p. 5 ~~“and therefore confound results across psychology and behavioural science~~

Replaced with:

“and therefore confound measures”

Removed from p.5 ~~other paradigms across behavioural science~~

Replaced with: “other paradigms”

Removed from p.27

~~“While these results are important for the interpretation of these and other closely related embodiment effects (e.g., full body illusions, Blanke & Metzinger, 2009), our intention is not to single out this field. Rather, these cases are intended as examples to draw attention to possible roles of phenomenological control effects across behavioural science.”~~

~~“is likely to”~~

The remainder of the text in this paragraph has been edited so that the claims are much more cautious and has been added to the end of an additional paragraph on p.30 (see below).

Removed from p. 27: ~~“Investigation of the role of phenomenological control therefore has the potential to transform human behavioural science”~~

Removed from p. 27: ~~“be necessary to test for similar relationships even in low level tasks. Consideration of phenomenological control may transform understanding of behavioural science.””~~

Removed from p.29: ~~“We do not know the extent of phenomenological control effects, but they cannot be ruled out in any study involving human report.”~~

Added to p.29:

“While here we report correlational evidence that trait hypnotisability (phenomenological control in the hypnotic context) predicts embodiment measures, we do not directly provide evidence for anything beyond these relationships. However, our inferences draw upon a vast literature investigating trait hypnotisability and response to imaginative suggestion and the proposal that these correlations indicate that response to the rubber hand illusion and mirror-sensory synaesthesia at least partially is a parsimonious theory (simple because it posits one mechanism to explain a wide range of phenomena) which cannot currently be ruled out, and which may have potentially far reaching implications. Any study in which the experimenter’s expectations are discernible to the participant and for which the generation of an experience could, in principle, influence measures may be affected by trait differences in phenomenological control. We predict that experiments which measure behaviour linked to relatively high-level cognitive processes will be most susceptible to the generation of experience in response to demand characteristics. For example, experimental paradigms in embodiment research are based on experimental designs which are closely related to the rubber hand illusion (e.g., the full body illusion) and we would expect to find similar relationships between hypnotisability and these effects. Similarly, given the classical suggestion effect of the experience of involuntariness, we should expect to find a relationship between reports of changes in the sense of agency (e.g., Wegner, 2003) and phenomenological control. Indeed, Cioffi, Banissy & Moore (2016) report that mirror-touch synaesthetes are particularly vulnerable to manipulations of the experience of agency. The high hypnotisability of mirror touch synaesthetes presents a simple explanation for this; one of the defining characteristics of high hypnotisables is an ability to generate alterations of the experience of agency. However, at this stage it cannot be ruled out that there may even be phenomenological control effects in apparently low level tasks, if those tasks are not as free of top-down influence as has been assumed (e.g., Lifshitz et al, 2013; Getz & Kubovy, 2018). Predictions of relationships between candidate measures can be straightforwardly tested using trait measures of phenomenological control, as in the present studies.”

5. - So far, I have commented on the fact that the researchers' claims seem to go too far beyond what is given in the data, indeed, one reason to suspect that these results may not generalize is that other sensory modalities tend to show much less susceptibility to the sorts of illusory phenomena explored here. There is a very active debate, for example, about whether vision and audition are subject to top-down phenomenological control in the first place (e.g., Dunning & Balci, 2013, Current Directions in Psychological Science; Firestone & Scholl, 2016, Behavioral and Brain Sciences; Lupyan, 2015, Review of Philosophy and Psychology; MacPherson, 2012, Philosophy and Phenomenological Research; Norris, McQueen, & Cutler, 2000, Behavioral and Brain Sciences; Proffitt, 2006, Perspectives on Psychological Science), with many arguing that such effects do not and cannot occur, and others arguing that always or often occur. I do not know of similar arguments in the tactile domain, because it's my sense that researchers tend to think that somatosensory phenomena are more subject to top-down influence. But if that's the case, then these results may not generalize as far as the authors imagine. A future version this paper would make a more powerful argument about perception as a whole if it better engaged with the broader literature on top-down effects on perception.

The reviewer is mistaken in thinking that the evidence presented here does not show top down effects on domains other than somatosensory arising from demand characteristics.

Consider RHI statement S4 – “The rubber hand began to resemble my own (real) hand, in terms of shape, skin tone, freckles, or some other visual feature.”

This was predicted by hypnotisability and therefore is an example of a demand characteristic driven visual hallucination. The figure below (figure 3c) shows the relationship between visual hallucination report and hypnotisability:

The reviewer is also mistaken that somatosensory phenomena are more subject to top-down influence than other domains. To the contrary, there is reliable evidence for top-down effects of imaginative suggestion in all modalities. Visual and auditory hallucinations are common suggestion effects (more common than tactile hallucinations) and form part of many hypnotisability scales, including the SWASH scale employed here; see Terhune, Cleeremans, Raz & Lynn, 2017 for a recent review of top down effects of imaginative suggestion). It is of course an empirical question whether the specific effects of imaginative suggestion (phenomenological control) are similar across other modalities. Here we present evidence that relationships are similar across, for example, touch and visual experience (S1, S2 and S4 RHI statements). We believe that exploring such relationships across other domains is an exciting opportunity for future research, motivated by the present study. The examples given by R2 of varying top down effects in perception had already attracted our attention, and we are conducting research to establish the degree to which trait phenomenological control can account for the variable evidence for top-down effects in psychological science (phenomenological control may be a 'hidden moderator', with top-down effects occurring on average when average trait phenomenological control is relatively high, but failing to occur when it is relatively low).

Added to p.27-28:

“An anonymous reviewer was concerned that these relationships might apply to somatosensory experience alone. While the effects reported here are related to

embodiment and involve tactile hallucinations, there is no reason to expect that such relationships will be limited to such cases. RHI statement S4 – “The rubber hand began to resemble my own (real) hand, in terms of shape, skin tone, freckles, or some other visual feature.” was here predicted by hypnotisability and therefore is an example of a demand characteristic driven visual hallucination. Furthermore, the hypnotisability scale with which reports of mirror touch and pain correlate here contains ten suggestions, and only one of these (a suggestion that a mosquito can be felt landing on the participant’s hand) involves tactile experience. Other suggestions on the scale include auditory and gustatory hallucinations, negative visual hallucination, limb paralysis, apparently involuntary arm movement and amnesia. Imaginative suggestion effects are therefore not limited to somatosensory experience, and potentially any experience which can be generated in response to imaginative suggestion in a hypnotic context could be generated to meet expectancies arising from demand characteristics (for a review of hypnotic suggestion effects see Woody & Barnier, 2008; for a review of top-down effects in imaginative recent suggestion see Terhune, Cleeremans, Raz & Lynn, 2017).

There is an active debate about top-down effects in psychological science (e.g., Firestone & Scholl, 2016), but evidence from imaginative suggestion research is rarely considered in contemporary discussions of this issue. Trait differences in phenomenological control potentially confound this literature, because the presence or absence of top-down effects is likely to vary according to the phenomenological abilities of the participants in a given sample. The ability to control phenomenology in response to imaginative suggestion is a stable, normally distributed trait and, whether or not it is measured, any participant in any psychology experiment will be drawn from this distribution. When phenomenological control is not measured, it may act as a hidden moderator in measures of experience. So, if a sample inadvertently over-represents those from the lower half of the distribution, top-down effects on perception will be much lower than if the sample over-represents those from the top half. Therefore, while demand characteristics may well account for reports of, for example, the weight of a back-pack affecting the perceived slope of an incline (Durgin et al, 2009), in some participants the reports would still reflect experience of a steeper slope when wearing a heavy backpack (through a creative act of imagination which is experienced as involuntary). Both the argument that the effects are due to demand characteristics and the argument that perception of the slope is altered would be correct (though of course any issues regarding generalizability of demand characteristic effects would hold). Consider the visual hallucination ‘control’ item in Study 2. The presented linear model predicts almost maximum agreement scores for maximum hypnotisability scores, but disagreement when hypnotisability scores are low. If we imagine two samples which are weighted either toward low or high phenomenological control, we would expect to find evidence for visual hallucination effects in one sample and evidence against such effects in the other. Each time the average phenomenological control ability is sufficiently high, an effect would be found, but when it is not, no effect would be found. This situation parallels that in evidence for top-down effects in psychology experiments (for example in the variability across studies of the effects of action constraints (such as wearing a heavy backpack) on distance perception; for a meta-analysis see Molto, Nalborczyk, Palluel-

Germain & Morgado, 2019). It is possible therefore that phenomenological control may be a hidden moderator driving variable results in psychology experiments and contributing to the ongoing ‘replication crisis’ (Chambers, 2019). Studies of top-down effects on perception are difficult to interpret without taking into account trait differences in phenomenological control.”

6. I'm sorry that this review has been mostly negative; I do have a positive impression of the work. But for me the conclusions the authors attach to it are just far too strong and broad to justify publication in this form.

We thank the reviewer for their close attention to the manuscript. We hope that our comments and revisions have assuaged their concerns.

Reviewers' Comments:

Reviewer #1:

Remarks to the Author:

This revision addresses my concerns adequately.

Reviewer #2:

Remarks to the Author:

My feelings about this paper are mostly unchanged. It is an important advance in our understanding of a circumscribed set of psychological phenomena, namely certain very unusual somatosensory illusions, including the RHI and mirror-touch synaesthesia. It does not seem to have the wide-ranging consequences that were previously attached to it, which are the sort of consequences that would make it attractive to readers of this very high profile and general interest journal.

The authors agreed that their initial claims were too strong, and so have walked back some of those claims. The paper no longer claims that these results will transform human behavioral science, but still suggests that psychology experiments of all stripes (not just unusual somatosensory illusions) need to accommodate these results, and still frames the results as the discovery that subjects can control their phenomenology in some kind of general way. This is true of the paper itself, and also the response. For example, the authors write that these results "need to be taken into account in the design of future experiments testing changes in experience". That is still an extremely strong claim, and it is one I remain very skeptical of. It is just not true, or at least has not been shown, that "future experiments testing changes in experience" (which I read as applying to any study about phenomenology) "need" to accommodate this new result showing that the strength of the rubber hand illusion is correlated with suggestibility. And I suspect they won't. I am just being sincere in doubting that this paper will make researchers who study "experience" of all kinds (vision, audition, olfaction, etc.) change how they do their work, because I suspect that they, like me, will interpret these results as applying mostly or only to the present context, ie fascinating but very unusual somatosensory illusions.

More generally, while it is appropriate and intellectually honest of the authors to reduce the strength of these claims (even though as I've just noted the authors still want to say that these results apply much more broadly than the context under study), I don't see how it's an approach that should make this paper more suitable for publication here. A different approach would have been to do more work, for example by exploring the role of hypnotisability in some new and very different experiential phenomena. If the authors truly want to demonstrate phenomenological control in a way that really does apply to all sorts of psychology experiments, the authors could have shown that hyponotisability also affects phenomenology in studies of visual or auditory phenomenology. If the authors had taken that approach, I would be more supportive of publication here at Nature Communications, since the results would become much more general, as originally intended and as still claimed to an extent. But since we're just talking about the same data interpreted more conservatively, I'm not seeing the strong case for publication here.

Additionally, the authors now make some new arguments that I'm afraid are not very compelling. For example, in response to my concern that these studies only apply to somatosensory phenomena, they write:

"Consider RHI statement S4 – "The rubber hand began to resemble my own (real) hand, in terms of shape, skin tone, freckles, or some other visual feature." This was predicted by hypnotisability and

therefore is an example of a demand characteristic driven visual hallucination. The figure below (figure 3c) shows the relationship between visual hallucination report and hypnotisability"

This does not rise to the standard of evidence of a visual hallucination. It is just ratings of agreement with a statement. I am not an expert on this issue, but it is hard for me to imagine that a vision journal would accept this kind of result as genuine evidence of a hallucination. It seems pretty clear what the "right" answer to this question is supposed to be, and so suggestibility could make subjects more likely to agree with this question, not more likely to hallucinate. The study of visual hallucinations uses much richer and more sensitive measures than rating scales reporting agreement with phrases like this. And if it wouldn't be acceptable in a proper vision context, it is hard to imagine how it could or should be acceptable for an even more general audience. This only strengthens my fear that the findings cannot be interpreted as broadly as the authors wish.

The authors also respond to the worry about control by stating that "no hypnosis researchers have yet taken issue with this terminology". I'm not sure the relevance of this. My worry is not about what hypnosis researchers will think, it is about what researchers who study phenomenology more generally will think. The paper is not about imaginative suggestion itself, it is about whether baseline differences in suggestibility affect the strength of somatosensory illusions. That is different, and doesn't demonstrate control in any strong sense I'm familiar with. Again this returns to my worry that the broader community these authors are speaking to (most relevantly the readership of Nature Communications) won't be swayed by these claims.

I'm sorry to write this way, but I just really think the authors are not accurately imagining how other researchers will hear these claims. For an analogy, imagine that a researcher who studies smell or taste discovered that individual differences in suggestibility changed how wine tasters reported the taste of wine when presented with adjectives others used to describe wine. Would we say that all other research on phenomenology needs to take this result into account in future studies? I think not. But that is precisely how many of the claims in this paper read, at least to me.

This review has mostly been negative, and I feel bad about that. This is a lovely series of studies reporting that a few well-known illusions are driven by factors that we previously didn't know about. Those who study these phenomena will surely take notice. But I continue to think that these results are much more limited in their reach than the authors are claiming them to be.

I will note that since my comments are mostly about the framing and not the experiments, I would be happy to defer to other reviewers or to the editor if I am in the minority here. If the editor or other reviewers do think that these results are of much more general interest than I do, such that my view is not the popular one, I would not be unhappy to see this paper published here. It is a great collection of results. I just think the authors are getting carried away with their interpretation, even in revision.

Reviewer #1 (Remarks to the Author):

This revision addresses my concerns adequately.

Reviewer #2 (Remarks to the Author):

My feelings about this paper are mostly unchanged. It is an important advance in our understanding of a circumscribed set of psychological phenomena, namely certain very unusual somatosensory illusions, including the RHI and mirror-touch synaesthesia. It does not seem to have the wide-ranging consequences that were previously attached to it, which are the sort of consequences that would make it attractive to readers of this very high profile and general interest journal.

The authors agreed that their initial claims were too strong, and so have walked back some of those claims. The paper no longer claims that these results will transform human behavioral science, but still suggests that psychology experiments of all stripes (not just unusual somatosensory illusions) need to accommodate these results, and still frames the results as the discovery that subjects can control their phenomenology in some kind of general way. This is true of the paper itself, and also the response. For example, the authors write that these results "need to be taken into account in the design of future experiments testing changes in experience". That is still an extremely strong claim, and it is one I remain very skeptical of. It is just not true, or at least has not been shown, that "future experiments testing changes in experience" (which I read as applying to any study about phenomenology) "need" to accommodate this new result showing that the strength of the rubber hand illusion is correlated with suggestibility. And I suspect they won't. I am just being sincere in doubting that this paper will make researchers who study "experience" of all kinds (vision, audition, olfaction, etc.) change how they do their work, because I suspect that they, like me, will interpret these results as applying mostly or only to the present context, ie fascinating but very unusual somatosensory illusions.

More generally, while it is appropriate and intellectually honest of the authors to reduce the strength of these claims (even though as I've just noted the authors still want to say that these results apply much more broadly than the context under study), I don't see how it's an approach that should make this paper more suitable for publication here. A different approach would have been to do more work, for example by exploring the role of hypnotisability in some new and very different experiential phenomena. If the authors truly want to demonstrate phenomenological control in a way that really does apply to all sorts of psychology experiments, the authors could have shown that hyponotisability also affects phenomenology in studies of visual or auditory phenomenology. If the authors had taken that approach, I would be more supportive of publication here at Nature Communications, since the results would become much more general, as originally intended and as still claimed to an extent. But since we're just talking about the same data interpreted more conservatively, I'm not seeing the strong case for publication here.

The suggestion of additional studies of phenomenological control in a range of experimental psychology contexts (those prominently involving subjective report) is well taken and is exactly what we hope our study will motivate. It would lie significantly beyond the present scope to undertake and report such studies here. However, we recognise that there is therefore a need to further clarify and 'tone down' our current claims. We have made a number of revisions throughout the paper for this purpose:

p. 5 Added: "These studies are intended as test cases which, because of commonality with phenomenological control effects have the greatest likelihood of providing evidence consistent with the theory. Of course, the extent to which different paradigms in psychology

produce effects due to phenomenological control is an open empirical question. Here we test predictions in mirror synaesthesia and the rubber hand illusion.”

p.5 deleted: If trait differences in phenomenological control can account at least partially for effects, other paradigms may produce genuine experiential and physiological change based only or partly on demand characteristics – a possibility which would substantially alter standard interpretations of results in these paradigms.

p.29 “It is possible therefore that phenomenological control may be a hidden moderator driving variable results in psychology experiments...”

Changed to:

“It is possible therefore that phenomenological control may be a hidden moderator driving variable results in some psychology experiments...”

p.29 ”Studies of top-down effects on perception are difficult to interpret without taking into account trait differences in phenomenological control.

Changed to:

“Studies of top-down effects on perception may usefully take into account trait differences in phenomenological control.”

p.29 deleted:

“and which may have potentially far reaching implications.. Any study in which the experimenter’s expectations are discernible to the participant and for which the generation of an experience could, in principle, influence measures may be affected by trait differences in phenomenological control.”

p.30 deleted: “However, at this stage it cannot be ruled out that there may even be phenomenological control effects in apparently low level tasks, if those tasks are not as free of top-down influence as has been assumed (e.g., Lifshitz et al, 2013; Getz & Kubovy, 2018). Predictions of relationships between candidate measures can be straightforwardly tested using trait measures of phenomenological control, as in the present studies.”

p.30 “...has major implications for research involving human participants.”

Changed to:

“This study motivates a systematic exploration of the influence of phenomenological control across a range of experimental contexts in which subjective phenomenological reports are prominent.”

Additionally, the authors now make some new arguments that I'm afraid are not very compelling. For example, in response to my concern that these studies only apply to somatosensory phenomena, they write:

"Consider RHI statement S4 – “The rubber hand began to resemble my own (real) hand, in terms of shape, skin tone, freckles, or some other visual feature.” This was predicted by hypnotisability and therefore is an example of a demand characteristic driven visual hallucination. The figure below (figure 3c) shows the relationship between visual hallucination report and hypnotisability"

This does not rise to the standard of evidence of a visual hallucination. It is just ratings of agreement with a statement. I am not an expert on this issue, but it is hard for me to imagine that a vision journal would accept this kind of result as genuine evidence of a hallucination. It seems pretty clear what the "right" answer to this question is supposed to be, and so suggestibility could make subjects more likely to agree with this question, not more likely to hallucinate. The study of visual hallucinations uses much richer and more sensitive measures than rating scales reporting agreement with phrases like this. And if it wouldn't be acceptable in a proper vision context, it is hard to imagine how it could or should be acceptable for an even more general audience. This only strengthens my fear that the findings cannot be interpreted as broadly as the authors wish.

p.27: “is an example of demand characteristic driven visual hallucination.”

Changed to “may be an example of demand characteristic driven subjective report compatible with a visual hallucination”

[Please note that none of our claims depend on the interpretation of S4 responses as reflecting perceptual visual hallucinations]

The authors also respond to the worry about control by stating that "no hypnosis researchers have yet taken issue with this terminology". I'm not sure the relevance of this. My worry is not about what hypnosis researchers will think, it is about what researchers who study phenomenology more generally will think. The paper is not about imaginative suggestion itself, it is about whether baseline differences in suggestibility affect the strength of somatosensory illusions. That is different, and doesn't demonstrate control in any strong sense I'm familiar with. Again this returns to my worry that the broader community these authors are speaking to (most relevantly the readership of Nature Communications) won't be swayed by these claims.

I'm sorry to write this way, but I just really think the authors are not accurately imagining how other researchers will hear these claims. For an analogy, imagine that a researcher who studies smell or taste discovered that individual differences in suggestibility changed how wine tasters reported the taste of wine when presented with adjectives others used to describe wine. Would we say that all other research on phenomenology needs to take this result into account in future studies? I think not. But that is precisely how many of the claims in this paper read, at least to me.

This review has mostly been negative, and I feel bad about that. This is a lovely series of studies

reporting that a few well-known illusions are driven by factors that we previously didn't know about. Those who study these phenomena will surely take notice. But I continue to think that these results are much more limited in their reach than the authors are claiming them to be.

I will note that since my comments are mostly about the framing and not the experiments, I would be happy to defer to other reviewers or to the editor if I am in the minority here. If the editor or other reviewers do think that these results are of much more general interest than I do, such that my view is not the popular one, I would not be unhappy to see this paper published here. It is a great collection of results. I just think the authors are getting carried away with their interpretation, even in revision.

We thank the reviewer for their detailed attention to this manuscript over multiple iterations, and for their kind description of our experiments.